# Technically Challenging Percutaneous Interventions of Chronic Total Occlusions Are Associated with Enhanced Platelet Activation

**DOI:** 10.3390/jcm12216829

**Published:** 2023-10-29

**Authors:** Ádám Illési, Zsolt Fejes, Marianna Pócsi, Ildikó Beke Debreceni, Katalin Hodosi, Béla Nagy Jr., János Kappelmayer, Zsolt Kőszegi, Zoltán Csanádi, Tibor Szük

**Affiliations:** 1Department of Cardiology, Faculty of Medicine, University of Debrecen, 4032 Debrecen, Hungary; illesi.adam@med.unideb.hu (Á.I.); koszegi@med.unideb.hu (Z.K.); kardiologia@med.unideb.hu (Z.C.); 2Doctoral School of Kálmán Laki, Faculty of Medicine, University of Debrecen, 4032 Debrecen, Hungary; 3Department of Laboratory Medicine, Faculty of Medicine, University of Debrecen, 4032 Debrecen, Hungary; fejes.zsolt@med.unideb.hu (Z.F.); pmarcsi89@gmail.com (M.P.); ideb@med.unideb.hu (I.B.D.); nagy.bela@med.unideb.hu (B.N.J.); kappelmayer@med.unideb.hu (J.K.); 4Institute of Internal Medicine, Faculty of Medicine, University of Debrecen, 4032 Debrecen, Hungary; hodosi@med.unideb.hu; 5András Jósa University Teaching Hospital, 4400 Nyiregyhaza, Hungary

**Keywords:** chronic total coronary occlusion, platelet activation, coronary intervention, microRNA, biomarker

## Abstract

Percutaneous coronary intervention (PCI) is a frequently performed treatment option for recanalization in patients with chronic total occlusion (CTO). As CTO-PCIs are often complicated and challenging for interventionalists, the stressful and damaging nature of the procedure can be remarkable, thus platelets can be easily activated. Our aim was to investigate the effect of CTO-PCI on platelet activation and the expression of selected circulating microRNAs (miR) of platelet and endothelium origin after CTO-PCI. In this study, 50 subjects after CTO-PCI were enrolled. Blood samples were obtained before PCI, at 2 days and 3–6 months after the procedure to measure the degree of platelet activation and the level of plasma miR-223, miR-181b, and miR-126. Patients were divided based on the characteristics of the intervention. Patients with higher Japanese CTO scores and longer duration of PCI showed significantly elevated platelet P-selectin positivity (*p* = 0.004 and *p* = 0.013, respectively) 2 days after the procedure compared to pre-PCI and increased concentration of soluble P-selectin 3–6 months after the intervention (higher Japanese CTO score: *p* = 0.028 and longer duration of PCI: *p* = 0.023) compared to baseline values. Shorter total stent length caused a significantly lower miR-181b expression at 3–6 months after the intervention (*p* = 0.031), while no difference was observed in miR-223 and miR-126. One stent thrombosis occurred during the follow-up period. Although these technically challenging CTO-PCIs may cause enhanced platelet activation right after the intervention and long-term endothelial cell dysfunction, these interventions are not associated with more adverse clinical events.

## 1. Introduction

Chronic total occlusion (CTO) is generally considered to be a complete blockage of the coronary arteries with Thrombolysis In Myocardial Infarction (TIMI) 0 flow with a duration of more than 3 months [1]. CTO is identified in approximately 20% of patients undergoing diagnostic coronary angiography [2]. CTOs can be solved by coronary artery bypass graft (CABG) surgery or percutaneous coronary intervention (PCI). Although PCI is considered a first-class choice in case of non-occlusive stenosis, less than 10% of patients with CTO receive a percutaneous treatment [3]. Interventional cardiologists take many factors into account when it comes to performing percutaneous revascularization. For estimating the expected difficulty of a CTO-PCI, a Japanese chronic total occlusion (J-CTO) score was created based on the following criteria: length of the occlusion, the status of the proximal cap (tapered or blunt), presence of angulation over 45° in the segment of CTO, presence of calcification and previous unsuccessful attempt for crossing the occlusion [4]. Based on all these, easy, intermediate, difficult, and very difficult (J-CTO scores of 0, 1, 2, and 3, respectively) procedures can be predicted. Nonetheless, in the last decade, significant improvement has occurred in the field of CTO-PCI: new intervention strategies, such as the retrograde wire approach [5,6] and the parallel wire technique [7] were involved, new devices were developed, and operator experience was also increased. These advances resulted in a tangible increment in the success rate of CTO-PCIs [8].

CTOs mainly evolve from progressive atherosclerosis, but can also develop as a consequence of an asymptomatic or untreated myocardial infarction [9]. While in acute or recent myocardial infarction the occluding thrombus is comparatively loose and not well-organized, a chronically occluded segment contains more collagen-rich fibrotic and calcified tissue [10], which often makes it challenging to cross the lesion with a guide wire. The first attempt for crossing the lesion is usually the wire escalation (WE) technique when the interventionalist drives the guide wire within the lumen of the vessel. If this intraluminal technique fails, subintimal dissection and reentry (DR) treatment can be attempted. During the DR procedure, the occlusion is bypassed by entering the subintimal space of the vessel wall. Both WE and DR technique is feasible with retrograde technique as well, where the distal part of the chronically occluded segment is approached from the direction of a collateral vessel [11]. Due to the complexity of the blockage, the longer duration, and the stressful nature of the intervention, the complication rate of CTO-PCIs is higher, than a PCI of a non-occlusive stenosis [12]. Despite the platelet activation depending on the patient’s individual characteristics [13]), these complications originate from an enhanced platelet, endothelial cell, and coagulation cascade activity, which are easily triggered by mechanical impacts of catheter intervention when crossing the occlusion, hence thrombosis and related complications can be expected after CTO-PCI [14]. 

A multitude of studies have demonstrated that platelets and coagulation cascade are activated during PCI [15,16,17]. Balloon dilation and stent deployment may destroy the endothelium lining the coronary arteries with stimulation of inflammatory cells, platelets, and the coagulation cascade. The injured endothelium is no longer thromboresistant, thus a layer of platelets and fibrin net are formed on this surface, which is followed by the generation of platelet-platelet, leukocyte-platelet and leukocyte-endothelial cell heterotypic aggregates. The adhesion of activated platelets to monocytes and neutrophils and the rolling of leukocytes on endothelial cells are mediated by P-selectin [18]. In addition, the catheters used during CTO-PCI can also trigger the activation of platelets [19], as well as the generated thrombin or the adenosine-diphosphate (ADP) released from red blood cells and platelets harmed by the balloon and stent [20]. These processes are fine-tuned by several additional factors one of them being the epigenetic regulator microRNAs (miRNA) which are short non-coding RNA molecules that play a key role in the posttranscriptional regulation of gene expression and protein translation that is required for numerous physiological and pathological cell functions [21]. Concerning platelet function, miR-223 can repress the expression of the P2Y12 receptor, which is one of the most important receptors of platelet function [22]. MiR-126 can also influence platelet activity, principally through the regulation of surface proteins—it also plays a role in the alteration of the expression of P2Y12 receptors [23]. MiR-126 can be expressed in endothelial cells as well, it participates in atherosclerotic processes [24]. Increased levels of miR-126 cause a decreased expression of vascular cell adhesion molecule-1 (VCAM-1) [25]. MiR-181b was associated with augmented plasma levels of VCAM-1 and E-selectin after coronary stenting [26], while its elevated level inhibited thrombin-induced endothelial cell activation [27]. Circulating miRNAs may act as suitable biomarkers in the laboratory investigation of severe clinical conditions [28,29]. A lot of clinical studies have investigated the safety of percutaneous interventions of CTOs in terms of long-term outcomes, but without the analysis of platelet-specific biomarkers. This gap in the literature prompted us to investigate the effect of PCI on platelet activation by measuring some platelet-related biomarkers.

## 2. Objectives

Our aim was to investigate the effect of CTO-PCI on platelet activation and the expression of selected miRNAs in relation to platelets and endothelial cells in patient groups divided based on different aspects. For this purpose, we determined the degree of platelet activation and measured some selected circulating miRNAs from plasma in two different post-PCI time points in comparison to baseline samples.

## 3. Patients and Methods

### 3.1. Patients

Consecutive patients (*n* = 59) who were referred to the Department of Cardiology and Cardiac Surgery, University of Debrecen for elective CTO-PCI between January 2019 and September 2020 were enrolled. Enrollment criteria included the age of over 18 years, CTO confirmed with coronarography, the presence of viable myocardial tissue distal from the occlusion, dual antiplatelet therapy prior to the intervention, and written informed consent from the patient. All patients were on a 100 mg of aspirin therapy due to confirmed coronary disease. There were a few patients, who had undergone stent implantation within half a year before the CTO-PCI, thus they arrived at the hospital with 100 mg of aspirin + 75 mg of clopidogrel antiplatelet therapy. Those patients with 100 mg of aspirin monotherapy received a 300 mg clopidogrel loading dose a day before the intervention. After all, every patient was on the same aspirin plus clopidogrel therapy, at the time of the first blood test. As the administration of antiplatelet agents was not changed after the intervention, during the follow-up period patients received the same antithrombotic therapy of 100 mg of aspirin plus 75 mg of clopidogrel. Exclusion criteria included unsuccessful attempt for revascularization, occurrence of severe life-threatening complications during the intervention, ongoing acute coronary syndrome, active malignant disease, and pregnancy. Diagnosis of CTO was claimed based on coronary angiography findings (TIMI flow 0). The duration of the occlusion was estimated by the onset of the first angina symptoms and patient history. Revascularization was considered successful if less than 30% of stenosis was left over in the treated vessel segment and TIMI flow 3 was restored. 

### 3.2. Blood Samples

Baseline anamnestic data and clinical and routine laboratory parameters were recorded at the admission to the clinic. Platelet activation-related biomarkers were quantified from blood samples which were obtained in three different time points. We measured the expression of surface P-selectin on platelets and the concentration of soluble P-selectin in plasma samples. To investigate the alteration of the miRNA levels of platelets, miRNA-223 and miRNA-126 quantitation was performed in parallel. MiRNA-181b levels were also measured as an additional miRNA regarding the endothelial cell function. To determine the effect of PCI on platelet activation, the study population was divided into two parts in the aspect of the intervention type (WE or DR), the J-CTO score (lower: 0 and 1 or higher: 2 and 3), the duration of the procedure (shorter: 40–117 min or longer: 118–255 min), and the total length of implanted stents (shorter: 16–53 mm or longer: 54–127 mm). The first blood sample was taken prior to the CTO-PCI, while the second and third samples were obtained 48 h and 3–6 months after the intervention, respectively. Peripheral blood samples were drawn into a Vacutainer® tube containing 3.2% Na-citrate (Becton Dickinson, San Jose, CA, USA) and after the measurement of platelet surface P-selectin by flow cytometry, whole blood was centrifuged at 1500× *g* for 15 min at room temperature for platelet-poor plasma separation. The samples were stored at −80 °C. At the end of the CTO-PCI, the duration and type (WE or DR) of the intervention, the number, size, and type of implanted stents, the number of guide wires, and the amount of heparin used were recorded. Every successful PCI ended with drug-eluting stent implantation. Some of the stents (Supraflex, Yukon Choice, Orsiro, Biomime Morph, Metafor) were sirolimus-eluting stents, while others (Promus, Synergy, Xience, Evermine) were coated with everolimus. In one-one cases, biolimus (Biomatrix Flex) and zotarolimus (Onyx) eluting stents were used as well. To assess the short- and midterm outcomes, subjects were regularly followed during outpatient visits or through telephone consultation. The incidence of the major adverse cardiovascular event (MACE) was also documented during the follow-up period (up to 12 months), this clinical endpoint included the incidence of stroke, myocardial infarction (MI), the target vessel revascularization (TVR), and death.

### 3.3. Laboratory Analyses

#### 3.3.1. Determination of Surface P-Selectin Expression on Platelets

Surface P-selectin expression was measured within 2 h from blood sampling from all three blood samples. Whole blood was fixed with 1% paraformaldehyde (PFA) and for staining CD42a FITC, CD62P, and Platelet Control IgG1 antibodies (Becton Dickinson) were used. For the investigation 10,000 platelets were analyzed from each sample and FC 500 flow cytometer (Beckman Coulter, Brea, CA, USA) was used for the measurements. 

#### 3.3.2. Determination of Soluble P-Selectin Concentrations

To perform enzyme-linked immunosorbent assay (ELISA), frozen plasma samples were further centrifuged at 10,000× *g* for 1 min at room temperature. Plasma levels of soluble P-selectin were determined with Quantikine ELISA Human P-selectin/CD62P Immunoassay ELISA kit (R&D Systems, Minneapolis, MN, USA) according to the manufacturer’s recommendations. 

#### 3.3.3. RT-qPCR Analysis of Extracellular miRNAs

Relative expression of cell-free miR-223, miR-126, and miR-181b was analyzed from peripheral plasma samples. Briefly, after being stored at −80 °C, thawed plasma samples were centrifuged at 10,000× *g* for 1 min at room temperature and 400 µL of cell-free supernatants were spiked-in with 5 pmol mirVana cel-miR-39 mimic (Ambion, Austin, TX, USA, ID:MC10956). Circulating miRNAs were then isolated with miRNeasy Kit (Qiagen, Hilden, Germany). The isolated total RNA was then reverse transcribed into cDNA using miRNA-specific stem-loop RT (reverse transcriptase) primer (500 nM, Integrated DNA Technologies, Leuven, Belgium) and TaqMan MicroRNA Reverse Transcription Kit (Applied Biosystems, Vilnius, Lithuania) according to the manufacturer’s instructions. Quantification of miRNAs was performed using RT-qPCR (real-time quantitative polymerase chain reaction), in which universal reverse primer (100 µM, Integrated DNA Technologies), miRNA-specific forward primer (100 µM, Integrated DNA Technologies), Universal Probe Library probe #21 (10 µM, Roche Diagnostics, Mannheim, Germany), Taq DNA polymerase (5 U/µL, Thermo Scientific, Vilnius, Lithuania) and deoxynucleotide (dNTP) Mix (2.5 mM, Thermo Scientific) were used. Each measurement was performed in triplicates using QuantStudio 12K Flex qPCR instrument (Applied Biosystems by Thermo Fisher Scientific, Waltham, MA, USA). To normalize the results of miRNA level, the *cel-miR-39* reference gene was measured in all the samples with the same RT-qPCR method. Oligonucleotides and RT-qPCR assays were designed by the software (version 3.4) developed by Czimmerer et al. [30], and sequences of primers that were used in this study are listed in Appendix A.

### 3.4. Statistical Analyses

Statistical analysis was performed using SPSS (IBM SPSS Statistics for Windows, Version 25.0., Armonk, NY, USA). Normality was determined using Kolmogorov–Smirnov test. Continuous variables with normal distribution were displayed as mean ± standard deviation (SD), and continuous variables with non-normal distribution were expressed as median and interquartile range (IQR). For comparisons, we used either the Wilcoxon or Mann–Whitney U test. Categorical data on patients’ clinical statuses during their follow-up were evaluated using the Chi2 test. Plots were constructed using GraphPad Prism statistical software (Version 8.0.1, San Diego, CA, USA). *p*-values < 0.05 were considered statistically significant. In the case of those patients (*n* = 4) in whom the fourth laboratory measurement was missed, the last observation carried forward (LOCF) method was applied and the last known result was used to replace the missing data.

### 3.5. Ethical Statement

Our study was approved by the Regional Scientific and Ethical Committee at the University of Debrecen, Clinical Center, and the National Scientific and Ethical Committee of Hungary (approval number: 6799-1/2019/EKU). Patients were enrolled in accordance with the Declaration of Helsinki. All participants gave written informed consent to be enrolled in this study.

## 4. Results

### 4.1. Baseline Characteristics of Patients

Out of 59 recruited patients, 50 subjects (84.7%) were treated successfully and thus participated in the study. The mean age ± SD was 61.78 ± 8.72 years, and 60% were male. The median follow-up period was 9 months. The relevant clinical and demographical data of the entire study population are shown in Table 1. Despite the difference in the age and administration of beta-blockers and statins that occurred in the J-CTO score division, we did not consider these parameters as relevant factors of platelet activation, thus the study subgroups were considered homogenous. J-CTO score and procedure duration divisions are shown in Appendix A, respectively.

### 4.2. Induced Expression of Surface P-Selectin after Long, Complicated Interventions

Patients with higher J-CTO (2–3) score values showed a significantly elevated platelet P-selectin positivity 2 days after the intervention (4.07 [3.18–7.17]% vs. 3.17 [2.48–4.22]%, *p* = 0.004) compared to baseline levels that was moderately decreased by 3–6 months (Figure 1B). In contrast, in the lower J-CTO (0–1) score group, no significant alteration was found in surface P-selectin expression 2 days and 3–6 months after the CTO-PCI (Figure 1A). Next, P-selectin positivity levels were analyzed based on the duration of the intervention. Accordingly, a shorter duration of PCI (40–117 min) resulted in no change in surface P-selectin expression (Figure 1C). When the procedure time was between 118–255 min, significantly elevated expression of P-selectin was measured 2 days after the CTO-PCI (3.64 [2.69–6.94]% vs. 3.16 [2.4–4.08]%, *p* = 0.013) compared to baseline values (Figure 1D). Interestingly, P-selectin expression stayed elevated by 3–6 months in those with a shorter duration of procedure, while these values were reduced in patients who underwent a longer intervention. Based on these results, we can conclude that platelets became more activated shortly after procedures with higher J-CTO scores and longer duration.

### 4.3. Elevated Post-PCI Soluble P-Selectin Concentrations at Higher J-CTO Scores and Longer PCI Duration

In the higher J-CTO score group, significantly elevated soluble P-selectin levels were measured 3–6 months after the intervention (54.2 [39.7–73.2] ng/mL vs. 45.8 [36–70.1] ng/mL, *p* = 0.028) compared to pre-intervention levels (Figure 2B). On the other hand, patients with lower J-CTO scores showed no alteration in the concentrations of soluble P-selectin after CTO-PCI (Figure 2A). No change was observed under relatively short procedures (Figure 2C), while patients who have undergone long intervention (118–255 min) showed significantly elevated soluble P-selectin levels 3–6 months after the PCI (54.8 [42.8–76.4] ng/mL vs. 49.5 [35.4–73] ng/mL, *p* = 0.023) compared to pre-PCI levels (Figure 2D). These data are in accordance with the surface P-selectin expression at 3–6 months in those individuals with longer and more complicated PCI suggesting that surface-bound P-selectin became soluble causing decreasing platelet P-selectin expression and higher plasma levels.

### 4.4. Altered Plasma microRNA Levels after CTO-PCI

In the WE group, no significant difference was found in the relative expression of miR-223 (Figure 3A), while a decreasing tendency occurred in the DR cohort 3–6 months after the CTO-PCI compared to baseline samples (0.012 [0.011–0.026] vs. 0.026 [0.017–0.045], *p* = 0.053, Figure 3B). Analyzing the results according to the three other categories, no significant alteration was found either in the 2 day or in the 3–6 month samples. The same can be claimed for miR-126, no significant change in the relative expression of miR-126 occurred between the post- and pre-PCI values in any categorization.

In the WE group, significantly lower expression of circulating miR-181b was detected after 3–6 months during the monitoring compared to the baseline values (0.023 [0.009–0.073] vs. 0.04 [0.014–0.104], *p* = 0.042, Figure 4A). In contrast, patients treated with the DR technique also showed reduced miR-181b levels throughout the study intervals, but did not reach a significant difference between two time points (Figure 4B). Shorter total stent length also resulted in a significantly lower expression of miR-181b at 3–6 months after the intervention (0.016 [0.007–0.049] vs. 0.037 [0.014–0.073], *p* = 0.031) compared to baseline values (Figure 4C). In the group with longer stents, no significant alteration was observed (Figure 4D). A nearly significant decrease occurred in the relative expression of miR-181b 3–6 months after the intervention (0.022 [0.011–0.053] vs. 0.034 [0.015–0.07], *p* = 0.068), when patients received shorter duration of treatment, compared to the baseline levels (Figure 4E). The longer duration of the procedure showed no significant alteration in relative miR-181b expression between pre- and post-PCI values (Figure 4F).

### 4.5. Clinical Follow-Up of Patients

During the median 9 months of the follow-up period, complication (stent thrombosis) was recorded in only one case (in the WE-treated group with lower J-CTO score and longer duration of PCI, 69 years old male, after 14 months) and no death occurred, therefore no significant change was documented in the complication rates between the study subgroups. Target vessel revascularization was performed in 4 cases, this ratio is the same as MACE data. No significant difference was observed in the MACE rate between the study groups.

## 5. Discussion

CTO lesions are technically challenging to deal with in current interventional cardiology. The complexity of the occluded segment often makes the procedure difficult: penetrating the proximal or distal cap with the guide wire, getting back into the intraluminal space in case of dissection and reentry technique, getting through a collateral vessel in a retrograde attempt, or reaching an adequate support guide wire position itself take much more expert skill and time. The stressful, complex, and prolonged interventions are associated with increased periprocedural complication rates [31], however, the appearance of ischemic adverse events is influenced by many factors such as mean platelet volume [32]. Thromboembolic factors are identified in numerous cases behind these complications. In the present study, significantly increased platelet activation was detected in the case of complicated and prolonged procedures 2 days after the CTO-PCI based on flow cytometric data. To our knowledge, no literature is available that investigates the effect of PCI on platelet activation in patients with CTO, data on this topic mainly deal with stable coronary artery disease population. Inoue et al. previously conducted a study on 22 patients with CAD who underwent PCI [33], the ratio of platelets positive for P-selectin (CD62P) was significantly increased after PCI compared to the pre-PCI values [33]. In the same study, patients who underwent only coronary angiography without stent implantation (*n* = 10) did not show a significant increase in the ratio of CD62 positive platelets after the intervention compared to baseline levels. Four years later, 48 stable angina patients were randomly assigned into either balloon angioplasty or coronary stent group (24–24 patients, respectively) by the same investigator team [34]. They found that patients in the stented group showed significantly increased transcardiac gradient (the value of coronary sinus blood minus the peripheral blood) of platelet surface expression of P-selectin immediately after the stent implantation, which increase was observed persistently [34]. Directly after balloon angioplasty, this increment was also detected, but at a lower level of significance and appeared to be transient during a 48-h observation period [34]. It should be also mentioned that aspirin + dipyridamol and aspirin + ticlopidine were administered before the interventions for the patients in these studies, respectively. Similar results were found by Nagy et al., who conducted a study on 25 stable angina patients who underwent elective PCI, in which expression of surface P-selectin of platelets was compared to values obtained from the subject (*n* = 20) who had diagnostic catheterization alone [35]. A significantly higher ratio of CD62P positive platelets was detected 15 min after PCI among stent-implanted patients compared to unstented subjects [35]. Data deriving from Inoue [33,34] and Nagy [35] are consistent with our results, as a higher level of platelet activation was identified shortly after the intervention if the procedure causes more vascular damage. 

Nagy et al. also conducted soluble biomarker measurements and did not identify any significant difference between stented and unstented patient groups when soluble P-selectin concentration was analyzed [35]. Inoue et al. investigated the soluble P-selectin values after coronary angioplasty among other circulating adhesion molecules [36]. They selected 25 patients with CAD who underwent elective PCI and measured the plasma levels of variable adhesion molecules from the coronary sinus and peripheral blood samples. The soluble P-selectin concentration measured from the coronary sinus was significantly elevated immediately after stent deployment compared to the pre-PCI values, and this difference increased further after 24 and 48 h [36]. Samples obtained from peripheral veins did not show any significant alteration in the concentrations of soluble P-selectin. Our study has detected a significant increment in the concentration of soluble P-selectin 3–6 months after the CTO-PCI if the procedure was complicated and prolonged. These data suggest that patients with higher J-CTO scores receive a more complex and longer intervention, which causes a greater extent of vascular damage. This harmful effect of PCI provokes more platelet adhesion and the production of more soluble P-selectin over time, to which endothelial cells could also contribute. The rapidly eliminated activated platelets are not in circulation for 3–6 months after the CTO-PCI, however, the soluble form of P-selectin is detectable even after a long period of time. 

There is no clinical trial available in the literature studying the connection between CTO patients and altered miRNA expression, thus, the present study is a novelty in this lesser-explored field. In a recent study, Landry et al. identified miR-223 as a potential repressor of mRNA encoding for the P2Y12 receptors [22]. Based on our findings, a downward tendency was observed in the relative expression of miR-223 only in the WE vs. DR procedure comparison. Patients treated with the DR technique showed a nearly significant decrease in the relative expression of miR-223 3–6 months after the intervention, while in the WE-treated group, no change was observed at the same time. The DR procedure is usually accompanied by considerable vascular injury due to entering the subintimal space. The observed change in the relative miR-223 expression might be in connection with the findings of Dai et al. [37], as they investigated the role of miR-223 in angiogenesis using ischemic cardiac microvascular endothelial cells (CMECs) from rats. They found that miR-223 inhibits the migration and proliferation processes of CMECs affecting the RPS6KB1/hif-1a signaling pathway, thus inhibiting the role of these cells in angiogenesis [37]. The nearly significant decrease in the relative miR-223 expression observed in the DR-treated group might be explained by the angiogenesis-inhibiting function of miR-223 since remodeling processes and stimulation of angiogenesis are more active after an intervention with extensive vascular injury. 

Evaluating the results obtained from miR-181b measurements we found a significant difference only in those patient groups, where the CTO-PCI was more simple, shorter duration, and less complicated. Patients who received WE intervention or shorter stents showed a significantly decreased relative miR-181b expression 3–6 months after the procedure compared to baseline levels. In the case of a lower J-CTO score and shorter duration of PCI, a decreasing tendency can be observed in the relative miR-181b expression, but the change did not reach the level of significance. As miR-181b can regulate the tumor necrosis factor-α (TNF-α) induced VCAM-1, intercellular adhesion molecule-1 (ICAM-1) and E-selectin expression [38], these data allow us to conclude, that relative expression of miR-181b could be decreased due to vascular inflammation caused by CTO-PCI. This assumption is also confirmed by the investigation performed by Fejes et al. [26]. In that study, reduced levels of miR-181b were detected, when in-stent restenosis, which is more common with the implantation of bare metal stents, occurred. Fejes et al. confirmed that in coronary endothelial cells in vitro, the inflammatory process caused by the invasive intervention was modeled with TNF-α treatment [26]. Evaluating these data, decreasing levels of miR-126 were expected by our research group in those interventions, which cause considerable vascular injury based on the findings by Harris et al. [25]; however, no significant difference was identified in relative miR-126 expression in neither categorization of the study population. The expression of these investigated miRNAs is usually decreased in those inflammatory conditions when platelets and endothelial cells become activated [23,24,27]. Despite the premedication of patients before and after PCI, platelet and endothelial cell dysfunction must have occurred after 3–6 months of both WE and DR interventions causing reduced miRNA levels. Accordingly, these miRNAs showed a higher sensitivity for cellular reactivity in contrast to differently altered levels of soluble P-selectin (Figure 1 and Figure 2) and VCAM-1 [39]. As both miRNAs have several targets to be fine-tuned, that is why there must be other consequences of abnormal protein production after PCI that we did not further investigate in this study. 

Our results suggest, that long, technically challenging CTO-PCIs are associated with increased platelet activation in an early- and increased endothelial cell activation in a late time frame, which may be the result of the more extensive vascular damage. Former data published by our research group also confirm the pivotal role of endothelial cells in the reparation processes following a stressful type of intervention, such as the DR procedure [39]. Enhanced platelet activation shortly after the procedure may predispose to a higher incidence of stent thrombosis, while increased endothelial activation can result in restenosis in the treated segment in the long term. These findings did not manifest in complications, however, further clinical events beyond the median 9-month follow-up period cannot be excluded. It should be mentioned, that control coronarography was performed only in a few cases, thus the incidence of restenosis could not be assessed. Based on the results regarding platelet activation, the administration of novel types of antiplatelet agents after a complex, challenging intervention in the early postoperative period should be considered. Complications appearing in the late follow-up period may be caused by the enhanced endothelial cell activity, thus our study can highlight the importance of research, which targets the attenuation of the activation of endothelial cells.

## 6. Conclusions

Long, complex, and technically challenging CTO-PCIs can cause a higher level of platelet activation shortly after the intervention, while the late postoperative period is characterized by increased endothelial cell activation. Despite these data, no difference was identified in adverse clinical outcomes in the 9-month follow-up period between patients with more simple and complicated procedures, thus both types of intervention can be used at similar safety levels.

## 7. Limitations

The relatively low number of cases is one of the most important limitations of our study. The relatively short follow-up period can also be considered as a limitation, which could lead to a low complication rate. The incidence of restenosis regarding the whole study population could not be assessed, as less than half of the patients had undergone control coronary angiography. Although the metabolism of clopidogrel is highly variable due to CYP2C19 allele variations [40], genotypical and pharmacogenetic analyses were not included in this research.

## Figures and Tables

**Figure 1 jcm-12-06829-f001:**
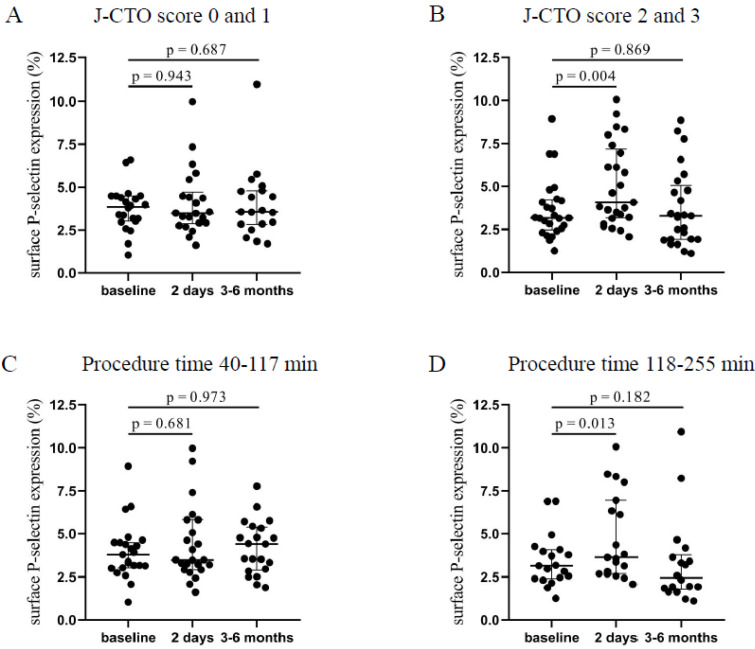
Comparison of surface P-selectin on platelets regarding the J-CTO score and the duration of the PCI. (**A**) Expression of surface P-selectin on platelets was measured before intervention (baseline), after 2 days, and at 3–6 months in the lower (0 and 1) and (**B**) higher (2 and 3) J-CTO score group. (**C**) Analysis of platelet surface P-selectin expression in patients treated with shorter (40–117 min) PCI at baseline, 2 days, and 3–6 months of intervention and (**D**) in patients received longer duration (118–255 min) of PCI. Wilcoxon test was used for comparisons. J-CTO: Japanese chronic total occlusion score. PCI: percutaneous coronary intervention.

**Figure 2 jcm-12-06829-f002:**
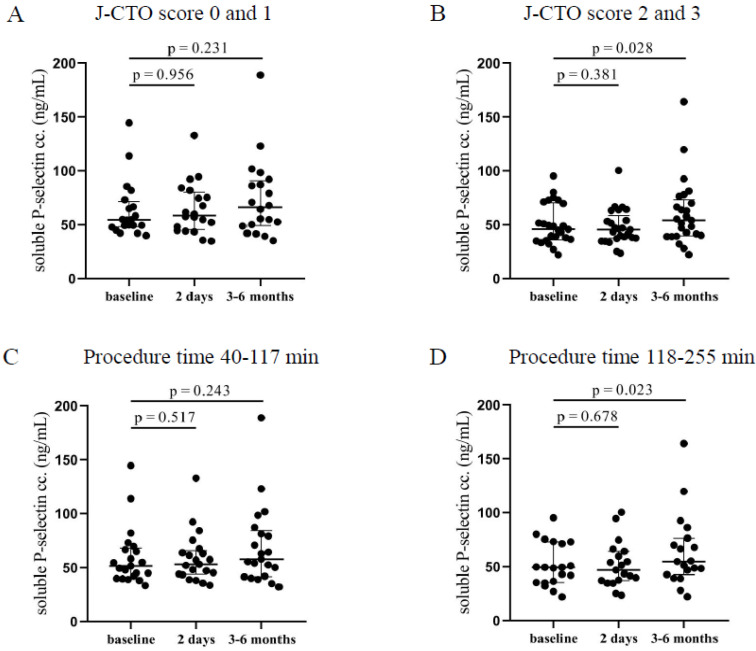
Comparison of soluble P-selectin concentrations in plasma regarding the J-CTO score and the duration of the PCI. (**A**) The concentration of soluble P-selectin in plasma was measured before intervention (baseline), after 2 days, and at 3–6 months in the lower and (**B**) higher J-CTO score group. (**C**) Analysis of soluble P-selectin concentrations in plasma in patients treated with shorter (40–117 min) PCI at baseline, 2 days, and 3–6 months of intervention and (**D**) in patients received longer duration (118–255 min) of PCI. Wilcoxon test was used for comparisons. J-CTO: Japanese chronic total occlusion score. PCI: percutaneous coronary intervention.

**Figure 3 jcm-12-06829-f003:**
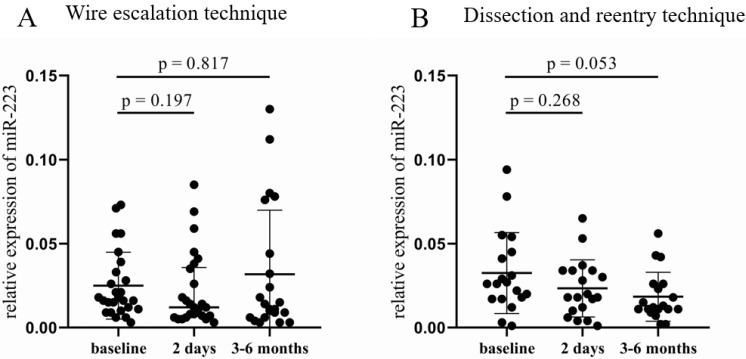
Comparison of relative miR-223 expression between the wire escalation and the dissection and reentry of treated patients. (**A**) Relative expression of miR-223 in patients treated with WE technique at baseline, 2 days, and 3–6 months after CTO-PCI. (**B**) Relative miR-223 expressions in patients who received DR procedure at baseline, 2 days, and 3–6 months after CTO-PCI. Wilcoxon test was used for comparisons. WE: wire escalation. DR: subintimal dissection and reentry. CTO: chronic total occlusion. PCI: percutaneous coronary intervention.

**Figure 4 jcm-12-06829-f004:**
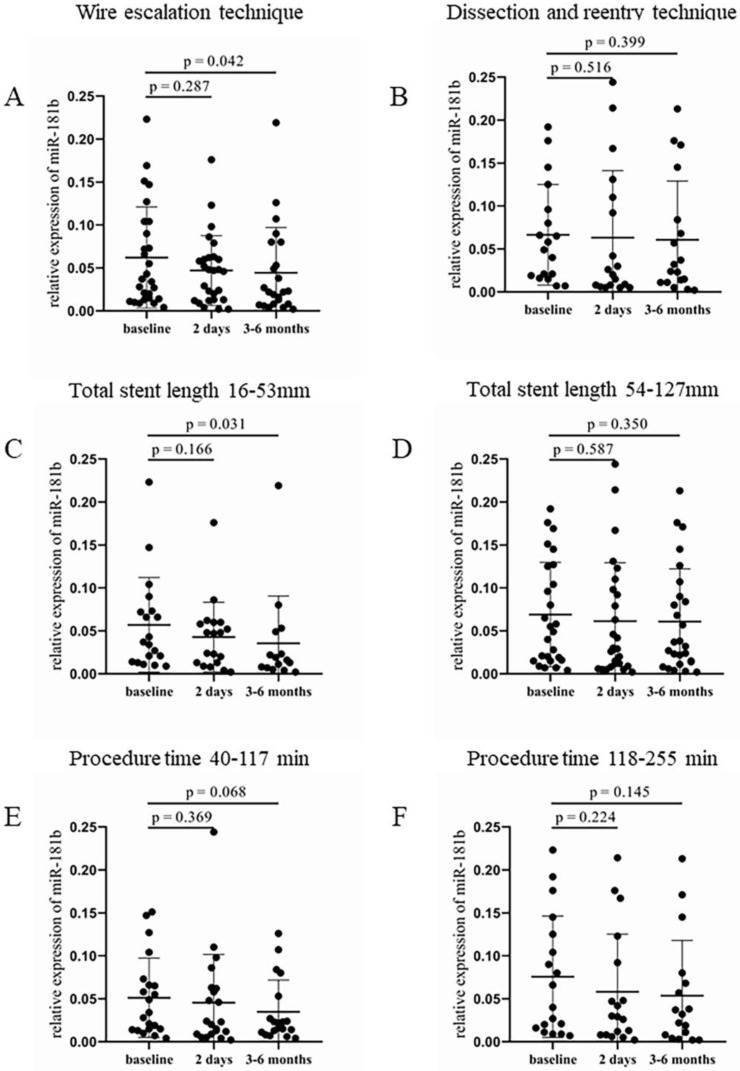
Comparison of relative miR-181b expression regarding the type of intervention, the total length of implanted stents, and the procedure duration. (**A**) Relative expression of miR-181b in patients treated with WE and (**B**) DR procedure at baseline, 2 days, and 3–6 months after the CTO-PCI. (**C**) Analysis of relative miR-181b expression in the case of shorter (16–53 mm) and (**D**) longer (54–127 mm) total stent length at baseline, 2 days, and 3–6 months after the intervention. (**E**) Relative miR-181b expression in patients received a shorter (40–117 min) and (**F**) longer (118–255 min) treatment at baseline, 2 days, and 3–6 months after the PCI. Wilcoxon test was used for comparisons. WE: wire escalation. DR: dissection and reentry. CTO: chronic total occlusion. PCI: percutaneous coronary intervention.

**Table 1 jcm-12-06829-t001:** Main demographic and clinical characteristics of recruited patients. Data are displayed as the mean ± standard deviation (SD). AMI: acute myocardial infarction, DM: diabetes mellitus, J-CTO: Japanese chronic total occlusion score, ACEi: angiotensin-converting enzyme inhibitor, ARB: angiotensin receptor blocker, and BB: beta-blocker.

Age (years)	61.78 ± 8.72
Male, *n* (%)	30 (60)
Hypertension, *n* (%)	42 (84)
Hyperlipidemia, *n* (%)	48 (96)
Obesity, *n* (%)	45 (90)
AMI in anamnesis, *n* (%)	22 (44)
DM in anamnesis, *n* (%)	19 (38)
J-CTO score	1.52 ± 0.95
ACEi/ARB, *n* (%)	44 (88)
BB, *n* (%)	45 (90)
Nitrates, *n* (%)	21 (42)
Statin, *n* (%)	42 (84)

## Data Availability

Data sharing is not applicable to this article.

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
