# Peer review of "Technically Challenging Percutaneous Interventions of Chronic Total Occlusions Are Associated with Enhanced Platelet Activation"

_jcm, 2023, doi:10.3390/jcm12216829_

Round 1

Reviewer 1 Report

Comments and Suggestions for Authors

The study is interesting but the key messages are not very novel. It is yet note that complex procedures have a major platelet activation increasing the ischemic risk. Moreover, the clinical impact of these analysis is very poor. Despite my opinion, I suggest some concerns to improve the quality of the paper:

- The authors should introduce a table with the antiplatelet therapy of the patients at the time of index procedure and during the followup. Underlining the types of loading and the maintenance doses.

- The authors should declare the types of implanted stents and their coated drugs.

- It would be interesting to know the lesion characteristic in particular the length and the number o mm of implanted stent. These characteristics reflect the ischemic risk related the complex lesions. The authors should introduce a table with these characteristic.

- A comparison with PRU (platelet reactivity unit) would be interesting.

- It would be interesting to know the genotyping and pharmacogenetic analysis.

- How do the authors explain the occurred of only one adverse clinical event despite the significantly elevated platelet P-selectin positivity in patients with complex lesions and long procedures?

Author Response

We would like to thank for the useful comments for Reviewer 1. Below we provide a point-by-point answer to all comments raised. The newly typed text highlighted in italic has also been inserted into the revised manuscript.

Comment #1: The authors should introduce a table with the antiplatelet therapy of the patients at the time of index procedure and during the followup. Underlining the types of loading and the maintenance doses.

Thank you for the useful comment, we accept this suggestion, and we have added a new paragraph into the manuscript (Lines 114-122). “Basically, all patients were on a 100 mg of aspirin therapy due to confirmed coronary disease. There were a few patients, who had undergone stent implantation within half a year before the CTO-PCI, thus they arrived at the hospital with 100 mg of aspirin + 75mg of clopidogrel antiplatelet therapy. Those patients with 100 mg of aspirin monotherapy received a 300 mg of clopidogrel loading dose a day before the intervention. After all, every patient was on the same aspirin plus clopidogrel therapy, at the time of the first blood test. As the administration of antiplatelet agents was not changed after the intervention, during the follow-up period patients received the same antithrombotic therapy of 100mg of aspirin plus 75mg of clopidogrel.”

Comment #2: The authors should declare the types of implanted stents and their coated drugs.

Thanks for the important comment. We have completed our methods with these details (Lines 150-154): “Some of the stents (Supraflex, Yukon Choice, Orsiro, Biomime Morph, Metafor) were sirolimus eluting stents, while others (Promus, Synergy, Xience, Evermine) were coated with everolimus. In one-one cases, biolimus (Biomatrix Flex) and zotarolimus (Onyx) eluting stents were used as well.”

Comment #3: It would be interesting to know the lesion characteristic in particular the length and the number o mm of implanted stent. These characteristics reflect the ischemic risk related the complex lesions. The authors should introduce a table with these characteristic.

Thanks for the advice. Based on the definition of CTO, every chronic occlusion counts as a complex lesion. We agree that complexity of the lesion has an influence on the ischemic risk. Even the Japanese-CTO score system makes a difference between occlusions shorter and longer than 20 mm. Obviously, a longer lesion makes the intervention more difficult, increasing the ischemic risk. In case of dissection and reentry procedure, the total stent length often exceeds the length of the lesion, depends on where it is possible to return the guidewire into the true lumen of the vessel. Below, we provide a table with the total stent length and total stent numbers in the lower J-CTO (0-1) and higher J-CTO (2-3) group division. Supplementary table 2 has been completed with additional data regarding the lesion characteristics.

J-CTO score 0 or 1

J-CTO score 2 or 3

number of patients

24

26

total number of stents

40

46

total stent length (mm)

1 178 mm

1 415 mm

average number of stents / PCI

1,67 mm

1,77

average total stent length / PCI

49,08 mm

54,42

Comment #4: A comparison with PRU (platelet reactivity unit) would be interesting.

Thanks for the great comment. We agree that this kind of comparison would be useful, as higher level of PRU is associated with adverse clinical events, however, PRU was not determined originally in these patients, and now we cannot produce it in the lack of special samples, retrospectively.

Comment #5: It would be interesting to know the genotyping and pharmacogenetic analysis.

Thank you for the interesting observation. “Although the metabolism of clopidogrel is highly variable due to CYP2C19 alleles variations [40], genotypical and pharmacogenetic analyses were not included in this research.” This has been mentioned in the Limitation section in lines 446-448.

Comment #6: How do the authors explain the occurred of only one adverse clinical event despite the significantly elevated platelet P-selectin positivity in patients with complex lesions and long procedures?

The registered one stent thrombosis was in the lower J-CTO and in the longer duration of PCI group. Although patients with a longer duration of PCI showed significantly elevated platelet P-selectin positivity, this alteration did not manifest at the level of complications eventually. This could be due to the relatively short follow-up period and limited number of cases.

Reviewer 2 Report

Comments and Suggestions for Authors

This article by Illesi et al aims to determine the impact of PCI prformed for chronic total coronary occlusion on endothelial health and function by measuring platelet activation and the level of plasma miR-223, miR-181b and miR-126 in a group of elderly individuals before the procedure and 2 days and 3-6 monhs after the procedure.

Some concerns are listed below:

Interestingly, P-selectin expression stayed elevated by 3-6 months in those with 224 shorter duration of procedure, while these values reduced longitudinally in patients who underwent a longer intervention”- please rephrase, as “longitudinal reduction” is unclear.

“Based on these results, we can conclude that platelets became more activated after procedures with higher J-CTO scores and longer duration.”- this does not seem to be a valid deduction. Please explain.

“In the WE group, significantly lower expression of circulating miR-181b was detected after 273 3–6 months during the monitoring compared to the baseline values (0.023 [0.009–0.073] vs  0.04 [0.014–0.104], p=0.042, Figure 4A).” Can you provide an explanation for this fact?

“It should be mentioned, that control coronarography was performed only in a few cases, thus incidence of restenosis could not be assessed”- please list this as a limitation. How many cases is “a few cases”? Was restenosis present in those patients? Were they symptomatic for restenosis? Were the others symptomatic for restenosis? Can you discuss the pattern of molecular chnages (miR-181b, miR-223 , soluble P-selectin concentrations)  in those specific individuals?

Author Response

We would like to thank for the important comments for Reviewer 2. Below we provide a point-by-point answer to all comments raised. The newly typed text highlighted in italic has also been inserted into the revised manuscript.

Comment #1 – „Interestingly, P-selectin expression stayed elevated by 3-6 months in those with shorter duration of procedure, while these values reduced longitudinally in patients who underwent a longer intervention”- please rephrase, as “longitudinal reduction” is unclear.

Thank you for your comment. The phrase „longitudinally” has been eliminated from the manuscript.

Comment #2 - “Based on these results, we can conclude that platelets became more activated after procedures with higher J-CTO scores and longer duration.”- this does not seem to be a valid deduction. Please explain.

Thank you for your opinion. We think that the significant elevation of platelet P-selectin positivity 2 days after a long and complicated procedure compared to pre-PCI values suggests that the characteristics of the intervention have a strong influence on platelet activation. The deduction above is also confirmed by the fact, that no significant change was occurred in the same parameter at the same time in case of more simple and shorter duration of PCI. However, a refinement has been placed in the cited paragraph, as this conclusion regards the early post-PCI period.

Comment #3 - “In the WE group, significantly lower expression of circulating miR-181b was detected after 3–6 months during the monitoring compared to the baseline values (0.023 [0.009–0.073] vs 0.04 [0.014–0.104], p=0.042, Figure 4A).” Can you provide an explanation for this fact?

Thank you for your excellent comment. We had originally analyzed the level of plasma miR-181b, miR-126 and miR-223 as potential biomarkers of endothelial cell and platelet activation [25, 37, 38]. “The expression of these miRNAs is usually decreased in those inflammatory conditions when these cell types become activated [23, 24, 27]. Despite the premedication of patients before and after PCI, platelet and endothelial cell dysfunction must have occurred after 3-6 months of both WE and DR interventions causing reducing miRNA levels. Accordingly, these miRNAs showed a higher sensitivity for cellular reactivity in contrast to differently altered levels of soluble P-selectin (Figures 1-2) and VCAM-1 [39]. As both miRNAs have several targets to be fine-tuned, that is why there must be other consequences of abnormal protein production after PCI that we did not further investigate in this study. Discussion was completed with the above in lines 409-417.

Comment #4 - “It should be mentioned, that control coronarography was performed only in a few cases, thus incidence of restenosis could not be assessed”- please list this as a limitation. How many cases is “a few cases”? Was restenosis present in those patients? Were they symptomatic for restenosis? Were the others symptomatic for restenosis? Can you discuss the pattern of molecular chnages (miR-181b, miR-223, soluble P-selectin concentrations) in those specific individuals?

Thank you for your advice. Information regarding the control coronarography were placed into the section ’Limitations’ in lines 444-446.

Control coronary angiography was performed in only 17 cases, of which 7 patients were symptomatic. Out of the seven symptomatic patients, stent thrombosis occurred in one case, in-stent restenosis was confirmed in 2 cases. The others were asymptomatic, and elective control coronarography was performed due to previously diagnosed stenosis on an other coronary artery.

Stent thrombosis: the soluble P-selectin concentration increased 2 days after the CTO-PCI and remained elevated 3-6 months later. As for miR-223, a minimal elevation in the relative expression was observed both in the 2 day and in the 3-6 months samples. Relative expression of miR-181b was increased considerably 2 days after the PCI, while a slightly elevation remained for the last blood sampling. This patient is from the lower (0-1) J-CTO, the shorter (0-53 mm) stent group, however the duration of PCI was long, despite the applied intraluminal (WE) technique.

Restenosis patient #1: concentration of soluble P-selectin showed a slightly decrease both in the 2 day and in the 3-6 months samples. The relative expression of miR-223 and miR-181b were decreased at the time of the third blood sampling and showed no change shortly after the intervention. This patient is from the higher (2-3) J-CTO, the longer (54-127mm) stent group, but the intervention was quite short, despite the dissection and reentry technique.

Restenosis patient #2: concentration of soluble P-selectin was slightly elevated 2 days after the CTO-PCI, and practically showed no change 3-6 months after the procedure. A mild decrease was observed in the relative expression of miR-223 2 days and 3-6 months after the intervention, while miR-181b levels did not change throughout the follow-up period. That patient had a quite complex lesion, the J-CTO score was 2. The patient belongs to the shorter stent length group, and received a shorter duration of PCI as WE technique was used.

After all, we had a low number of complications, and these individuals showed variable patterns of the examined parameters during the study intervals. Maybe the decrease in relative expression of miR-223 in case of patients with restenosis can be explained with the supposed angiogenesis inhibiting role of the miR-223. The decrease in the miR-181 level at one of the restenotic patients may suggest that the CTO-PCI was damaging and harmful, which was followed by vascular inflammation mediated by miR-181b as well.

Reviewer 3 Report

Comments and Suggestions for Authors

I have reviewed the manuscript entitled 'Technically challenging percutaneous interventions of chronic 2 total occlusions are associated with enhanced platelet activation'. 

High platelet activity can be due patient's individual characteristics. This should also be emphasized citing'The first six-month clinical outcomes and risk factors associated with high on-treatment platelet reactivity of clopidogrel in patients undergoing coronary interventions'.

The platelet volume is also very important which is found to be related with adverse ischemic events. Please briefly mention this issue in the discussion section citing 'Predictors of left atrial thrombus in acute ischemic stroke patients without atrial fibrillation: A single-center cross-sectional study'

Comments on the Quality of English Language

I have reviewed the manuscript entitled 'Technically challenging percutaneous interventions of chronic 2 total occlusions are associated with enhanced platelet activation'. 

High platelet activity can be due patient's individual characteristics. This should also be emphasized citing'The first six-month clinical outcomes and risk factors associated with high on-treatment platelet reactivity of clopidogrel in patients undergoing coronary interventions'.

The platelet volume is also very important which is found to be related with adverse ischemic events. Please briefly mention this issue in the discussion section citing 'Predictors of left atrial thrombus in acute ischemic stroke patients without atrial fibrillation: A single-center cross-sectional study'

Author Response

We would like to thank for the useful comments. Below we provide a point-by-point answer to all comments raised. The newly typed text highlighted in italic has also been inserted into the revised manuscript.

Comment #1 - High platelet activity can be due patient's individual characteristics. This should also be emphasized citing 'The first six-month clinical outcomes and risk factors associated with high on-treatment platelet reactivity of clopidogrel in patients undergoing coronary interventions'.

Thank you for the observation. We accept this suggestion, and an additional sentence regarding the patient’s individual characteristics was insterted to the Introduction in lines 67-68.

Besides the fact, that platelet activation also depends on the patient’s individual charac-teristics [13]), these complications are originated from an enhanced platelet, endothelial cell and coagulation cascade activity, which are easily triggered by mechanical impacts of catheter intervention when crossing the occlusion, hence thrombosis and related compli-cations can be expected after CTO-PCI [14].

The platelet volume is also very important which is found to be related with adverse ischemic events. Please briefly mention this issue in the discussion section citing 'Predictors of left atrial thrombus in acute ischemic stroke patients without atrial fibrillation: A single-center cross-sectional study'

Thank you for the suggestion, The important role of mean platelet volume has been mentioned in lines 326-327.

The stressful, complex and prolonged interventions are associated with increased periprocedural complication rates [31], however appearance of ischemic adverse events are influenced by many factors such as mean platelet volume [32].

Round 2

Reviewer 1 Report

Comments and Suggestions for Authors

The authors addressed the revision comments modifying the paper

I have not other comments to submit. 

Reviewer 2 Report

Comments and Suggestions for Authors

The authors have adressed my concern. Thank you!